# Comparison of Flexible Ureteroscope Performance between Reusable and Single-Use Models

**DOI:** 10.3390/jcm12031093

**Published:** 2023-01-30

**Authors:** Marius Bragaru, Razvan Multescu, Petrisor Geavlete, Razvan Popescu, Bogdan Geavlete

**Affiliations:** 1Department 3—Nephrology, Urology, Immunology and Transplant Immunology, Dermatology, Allergology, “Carol Davila” University of Medicine and Pharmacy, 050474 Bucharest, Romania; 2Department of Urology, “St. John” Emergency Clinical Hospital, 042122 Bucharest, Romania

**Keywords:** flexible ureteroscopy, reusable ureterscopes, single-use ureteroscopes, Olympus URF-V, UscopeUE3022

## Abstract

Background: Single-use flexible ureteroscopes for urinary retention have been developed in recent years as an alternative to reusable ureteroscopes in order to eliminate the risk of cross-infections and to solve the primary limitations of traditional reusable flexible ureteroscopes for urinary retention. Methods: In this study, we evaluated and contrasted three of the most recent types of flexible ureteroscopes, including two digital reusable versions (Olympus URF-V and Olympus URF-V2) and one single-use model (Pusen Medi-calUscope UE3022), in both ex vivo and in vivo scenarios. The influence of a variety of instruments on the flow of irrigation and its deflection was investigated ex vivo. In the in vivo investigation, a total of 40 patients were treated with retrograde fURS utilizing URF-V, 20 patients were treated with URF-V2, and 20 patients were treated with single-use fURS. The visibility and maneuverability of each fURS were evaluated by the same urologist during the procedures, and the results were compared. Results: Intraoperatively, we compared the image quality of reusable (URF-V and URF-V2) and single-use fURS USCOPE UE3022 cameras and found that there was no statistically significant difference between the two types of camera. The score for maneuverability was the same (4.2) regardless of whether we used the UscopeUE3022 or the URF-V2, but it was significantly lower (3.8, *p* = 0.03) when we utilized the URF-V. Irrigation was about the same when utilizing reused scopes, whereas employing a single-use scope was more than fifty percent more effective. Conclusions: The findings of our research indicate that reusable and single-use fURs have visibility and maneuverability characteristics that are at least comparable to one another. The possibilities of the single-use type in terms of irrigation flow and deflection are superior.

## 1. Introduction

The flexible ureteroscope (fURS) represents a very useful diagnosis and treatment instrument for upper urinary tract pathology. Hugh Hampton Young and RW MacKay performed the first known ureteroscopy (URS) in 1912 while utilizing a 9.5 Fr pediatric cystoscope [1]. The first ureteroscope was created and put into service in 1964 thanks to technological developments such as fiber optics [2]. This method has overtaken other urological procedures such as percutaneous nephrolithotomy in terms of its significance in the treatment of renal lithiasis. This is due to the ongoing development of technology, which has allowed it to become more effective in this regard. The use of percutaneous nephrolithotomy (PCNL) has become increasingly restricted as a result of the improvement of flexible ureteroscopes. This is especially due to the higher complication rate that PCNL presents, particularly in terms of life-threatening complications such as bleeding or urothorax [3]. Due to the lack of a working channel and active deflection capabilities, the early flexible endoscope had very limited functionality. However, when developments were achieved, ureteroscopes gradually developed the capacity for irrigation and active deflection, shifting from being solely diagnostic tools to also being therapeutic tools [4]. Digital flexible ureteroscopy was finally made possible in 2006 with the launch of the Olympus Invisio DUR-D thanks to the development of digital image sensors (Gyrus ACMI) [5,6].

Reusable digital flexible ureteroscopes have now been embraced by an expanding number of urologists [7]. However, using reusable flexible ureteroscopes comes at a significant financial cost. An initial investment of about $25,000 USD is required, in addition to expenses for the video processor and viewing device [5]. Even though reusable digital ureteroscopes are more durable than fiberoptic devices, this is nevertheless a cause for concern [8,9]. Recurrent damage that develops after each use might cause issues such as deflection loss and ultimately impair the function of the scope for subsequent patients [10]. After they are purchased, the costs for reprocessing, upkeep, and repairs can reach $90,000 to $100,000 annually [11,12]. Despite the efficiency of flexible ureteroscopy (fURS), its widespread adoption, especially in poorer nations, is hampered by its high ongoing expenses and the issue of reusable ureteroscopes’ poor durability [7].

Sterility is a major issue when using reusable flexible ureteroscopes [10,13]. According to a study by Ofstead et al., contamination (by bacteria, adenosine triphosphate, hemoglobin, and/or protein) could still be identified in the tested reusable ureteroscopes even after they were manually cleaned and sterilized using hydrogen peroxide gas [13]. Single-use flexible ureteroscopes have just become a reality as a solution to the problems associated with reusable ureteroscopes. These disposable ureteroscopes were recently developed by numerous companies as a result of ongoing innovation.

Flexible ureteroscopy can now treat renal calculi regardless of the calyx groups. In recent years, single-use fURS were developed as an alternative to reusable ureteroscopes in order to remove another unsolved weak point: the risk of cross-infection [14]. There are many single-use digital fURS already on the market. Single-use fURS are designed with the goal of alleviating the primary limitations of traditional reusable fURS, which include the high costs of acquisition and maintenance, the high risk of breakages, and the need for reprocessing [15]. 

This study aimed to compare two of the most recent flexible reusable ureteroscopes to a single-use ureteroscope in a variety of scenarios, including intraoperatively as well as ex vivo, making use of objective as well as subjective factors.

## 2. Material and Methods

We studied three of the most recent types of flexible ureteroscopes, all of which are popular in our area: two reusable digital models (the Olympus URF-V and the Olympus URF-V2) and one model that is intended for a single patient (Pusen MedicalUscope UE3022). To get as close as possible to real working conditions, the reusable scope was preused but checked before the investigation to ensure unimpaired functionality. The statistical analysis for our study was performed using a paired Student’s *t*-test.

### 2.1. Operative Technique

The procedures were performed with the patient in the standard lithotomy position, under spinal anesthesia. After cystoscopic identification of ureteral orifices, a 0.035” stiff hydrophilic guidewire is placed into the renal cavities under fluoroscopic control. A Cook Flexor 10.7/12 F was used in all cases. Initially, a thorough inspection of the upper, middle, and inferior calyx was performed. After we identified the stone, Holmium laser lithotripsy using a 270 µm fiber was performed in fragmentation mode, dusting, or both. When necessary, baskets were used for stone retrieval. At the end of the procedure, another inspection of all the caliceal groups was performed. The urologist who carried out the procedures was tasked with scoring the maneuverability during the two examinations of all of the caliceal groups. This was done in order to evaluate not only how successful the procedures were but also how easy it was to access all of the calyces. A JJ stent was indwelled at the end of the procedure.

### 2.2. Description of Physical Characteristics of Flexible Ureteroscopes (UscopeUE3022, URF-V, and URF-V2)

The main technical features of the instruments are summarized in Table 1. 

Olympus URF-V has an 8.5 Fr outer diameter, 9.9 Fr outer diameter insertion tube, a 670 mm working length, a standard 3.6 Fr working channel, and 180° angulation up/270° angulation down.

Olympus URF-V2 has an 8.4 Fr 670 mm length shaft and it is stiffer than that of its predecessor, aiming for easier access to the kidney. It also has a standard 3.6 Fr working channel and a 275° bi-directional angulation. Both reusable ureteroscopes have an improved insertion tube rotation function in order to minimize surgeon fatigue. Theoretically this improves the ergonomics, by allowing the surgeon to hold the endoscope in a neutral position and rotate the insertion tube independently. 

UscopeUE3022 is a single-use LED-lit, CMOS digital fURs. This model has a 9.2 Fr distal tip, a 650 mm length shaft, a standard 3.6 FR working channel for irrigation and insertion of instruments, and 270° bi-directional deflection.

### 2.3. Ex Vivo Comparison of Flexible Ureteroscopes (Uscope UE3022, URF-V, and URF-V2)

In an ex vivo scenario, irrigation flow and deflection loss were studied upon insertion of supplementary instruments into the working channel.

The irrigation flow and maximal deflection in each of the three ureteroscopes were measured with the working channel emptied out and with the following accessory instruments in place: a 0.035 inch guidewire, a 2 F ZeroTip basket, and a 270 micron Ho:YAG laser fiber. These measurements were taken with the ureteroscopes in the same orientation. In each of these tests, the ureteroscope was held in a perfectly straight position so that any fluctuations in flow or deflection that could have been caused by a working segment that was curved could be eliminated.

For the irrigation flow measurements, saline was placed at 150 cm above the endoscope, the same setting that we use in the operating room. The flow was measured with the working segment of the fURS in the straight position, initially with an empty channel and later with various instruments occupying the channel in order to get a more accurate reading. In total, three separate sets of measurements were taken. In the end, we decided to utilize the mean value.

The deflection angle was measured between the tangents to the straight working segment and the deflected tip, with a protractor on a photograph of the ureteroscope superposed onto millimetric paper.

The ability of each fURS to deflect was tested in a variety of configurations, beginning with the working channel left empty (Figure 1) and continuing with the channel filled with a variety of instruments, including a laser fiber measuring 270 m in diameter, a 2 Fr basket, and a guidewire measuring 0.035 inches. In each configuration, the maximum deflection was measured in both directions.

### 2.4. In Vivo Assessment of Flexible Ureteroscopes (Uscope UE3022, URF-V, and URF-V2)

The current study comprised a total of sixty patients who were diagnosed with lithiasis throughout the course of the previous three years. Of these, 20 consecutive patients received single-use fURS, 20 consecutive patients received retrograde fURS using URF-V, and 20 consecutive patients received URF-V2. 

The same urologist, who has considerable experience in the field of flexible ureteroscopy, carried out the intraoperative comparisons of the flexible ureteroscopes. At the conclusion of each procedure, the surgeon graded the visibility and maneuverability of the fURS on a scale from one to five, with one meaning “awful,” two meaning “poor,” three meaning “fair,” four meaning “good,” and five meaning “very good.”

In order to eliminate the possibility of bias in any of the 60 cases, a ureteral access sheath was utilized in each one. During the procedures, the performance of each fURS was recorded and analyzed so that its strengths and weaknesses could be contrasted. In addition, as highly essential parameters, the deflection loss, as well as the damage to the optical system, and the longevity of the equipment, were recorded.

## 3. Results

In regard to the acquisition costs, in our region, the reusable ureteroscope is 19-fold more expensive than the reusable one. The exploitation costs are more difficult to assess.

Intraoperatively, we discovered that there was no statistically significant difference in picture quality between reusable (URF-V and URF-V2) and single-use fURS USCOPE UE3022 (visibility score of 4.8, 4.7, and 4.8, *p* > 0.4). This was the conclusion we came to after comparing the three different scores (Table 2). 

Maneuvrability score was similar when we used UscopeUE3022 and URF-V2 (4.2) and significantly less when we used URF-V(3.8, *p* = 0.03) (Table 2).

Irrigation was similar on the reusable scopes and more than 50% improved when using a single-use scope. This difference was recorded both with an empty working channel and with the fiber laser inserted through it (Table 3).

When inserting a 270 micron Holmium laser fiber, deflection loss was 13% for URF-V2 and 8.7% for URF-V (significantly lower, *p* < 0.07) (Figure 2), while for the single-use scope the deflection remained relatively unchanged (statistically significant difference by comparison to the reusable scopes, *p* < 0.05). When the Zero-tip basket was used, deflection loss was similar for URF-V and URF-V2 (6.8% vs. 8.6%, *p* = 0.6) and significantly lower for USCOPE 3022 (2.2%, *p* < 0.01) (Table 4).

## 4. Discussion

Several studies have shown that reusable ureteroscopes suffer damage, requiring repair after 10 to 20 procedures [16]. Technological progress made this lifespan become much longer [17]. Disposable fURS have been developed to improve some of the unfavorable features reusable scopes may have as their availability, sterilization, or costly repairs [10].

In the last years, many disposable scopes have been developed, but not all have been adequately studied and compared. Comparison studies involving different kinds of flexible ureteroscopes are particularly helpful in characterizing changes in irrigation flow rates or deflection that might occur between manufacturers when working channels are either empty or occupied. The primary useful conclusion of these measurements is to anticipate the performance in vivo, which helps contribute to an ideal selection of situations. The three scopes we selected have not been properly compared and tested based on available literature.

Marchini et al. [18] conducted one of the earliest in vitro evaluations of UscopeUE3022by, whereas Salvado et al. [19] conducted an intraoperative evaluation and reported stone-free rates of up to 95% in 71 patients with a mean stone size of 11.4 mm. Both groups performed their evaluations simultaneously. In 2017, Johnston et al. conducted a prospective cohort study to assess the UscopeUE3022 ease of insertion, deflection, image quality, maneuverability, and overall performance, with 56 procedures performed in 11 international centers. UscopeUE3022 performed well with regard to maneuverability, deflection, and limb fatigue [20].

Kam J. et al. conducted a prospective comparative study that included 31 patients who underwent retrograde fURS using UscopeUE 3022 versus 64 patients using Olympus URF-V2. The URF-V2 group scored higher regarding visibility and maneuverability compared toUscope 3022 [16].

Our study demonstrated that all three models have particular advantages and disadvantages. Single-use fURS have the clear advantage of not requiring any sterilization process. They are similar to reusable scopes in terms of maneuverability, quality of vision, and clinical efficacy. Other advantages include avoiding the expensive reprocessing and repair costs.

Predictably, the diameter of the instrument proportionally influences the reduction of irrigation flow, although there are some variations depending on the flexible ureteroscope model.

However, perhaps surprisingly, at the same diameter of the irrigation canal, the disposable one has a higher flow, both with the empty canal and with it occupied by various instruments.

When using more extensive accessories, irrigation flow is practically canceled, but this does not have a real impact in most cases, with irrigation usually being more critical during lithotripsy than during the extraction phase.

Using a disposable tool, the previous reflexes employed during flexible ureteroscopy (such as reducing working time in the lower pole and relocating stones) are no longer necessary: you do not have to protect a very expensive device because it will not be used later.

All three models demonstrated good maneuverability. The manufacturers used the so-called excessive deflection in their latest models: a single of at least 270°, which appears to be easier to handle, offering improved access to the pyelocaliceal system.

However, the main debatable issue regarding single-use fURS remains the costs.

There is some overlap between the price ranges for disposable and reusable scopes [15]. Reusable fURS may be used for a long period, but everyone must keep in mind that they also involve maintenance costs influenced by a combination of factors: the number of surgeons who have access to the scope, the degree of training of the personnel involved in processing the scopes, the person who repairs it (the manufacturer or outsource vendors), etc. In some situations, even a new reusable fURS model may be used for an average of 21 procedures before requiring repair [15]. Moreover, supplementary costs come from the higher rate of urinary tract infections, total operative time, etc. Furthermore, over time degradation of the reusable ureteroscopes (the so-called “fatigability”) can cause inconsistent performance, and their damage can lead to substantial additional costs. When compared with the cost of re-fURS, the one-time cost of single-use fURS is still seen as being quite expensive. This is primarily because of the absence of any significant competitors in the market as well as their small market share [21].

In this particular investigation, only one endoscope of each reusable model was put through its paces. We did not take into account the stone-free rate or postoperative complications in our analysis. This is one of the limitations of the evaluation, along with the small sample size, the number of trials, and the fact that reusable ureteroscopes were not brand new at any point over the course of the research. The fact that several parameters were judged based on the subjectivity of the researcher is likely another source of bias. Our findings need to be validated by further testing and implementation in clinical settings.

## 5. Future Directions

Ureteroscopy is typically involved in the treatment of kidney stones as well as the management of other disorders related to the urinary tract. Concerns regarding the cost–benefit ratio of reusable ureteroscopes have precluded their widespread usage and spurred the development of single-use ureteroscopes. According to the data that is currently accessible, these innovative medications show promise as a potential treatment for disorders other than stones.

Previous studies constantly came to the conclusion that conventional fiber-optic devices could not compete with the performance and image quality of single-use ureteroscopes [22,23,24].

When the same qualities were contrasted with digital flexible ureteroscopes, however, differences between studies were found 6,8–10. This variation suggests that more investigation is required to definitively establish the comparability of single-use ureteroscope image quality and function. Additionally, the layout of the aforementioned investigations restricts the relevance of these findings in an in vivo human model [23,25,26,27].

Additionally, contradictions about the potential financial advantages of single-use ureteroscopes have been found. The average lifespan of reusable ureteroscopes after an initial repair, as well as the cost of reusable ureteroscopes per case, have shown significant variation in previous studies [8,9,27,28].

It is important to keep in mind the effects that utilizing a variety of tools and approaches can have on the surrounding natural environment. According to a study conducted by Baboudjian M., when one compares the disinfection and reprocessing of reusable endoscopes with the full lifespan of single-use endoscopes, the latter is associated with a reduction of at least 33% in the category of climate change, 50% in the category of the depletion of mineral resources, 51% in the category of ecotoxicity, 71% in the category of acidification, and 49% in the category of eutrophication et al. [29].

Recent studies about fluoroscopic exposure revealed similar results regarding stone-free rate and complication rate for ureteroscopy, but there were some differences in the operative time registered [30]. Shortening the procedure time represents an important aspect of flexible devices. A future study for comparing the impact of radiologic exposure in reusable fURS and single-use fURS may be useful to evaluate their characteristics.

There is also a lack of agreement among experts concerning the moment at which the two devices will be economically neutral. In point of fact, the financial value of single-use ureteroscopes seems to be highly dependent on the beginning expenses that are particular to the institution as well as the quantity of patient volume that is seen at care facilities. As a result, it might be to the advantage of care facilities to do their own cost-benefit analyses, taking into consideration the specific beginning expenses they incurred as well as the typical number of treatments they carried out.

## 6. Conclusions

According to the findings of our research, reusable and single-use fURs are at least comparable to one another in terms of their visibility and their capacity to be maneuvered. The single-use approach appears to have superior capabilities in terms of irrigation flow and deflection. It also has the potential to supply supplemental resources when dealing with challenging circumstances. These findings provide grounds for cautious optimism for the future by demonstrating the viability of multiple potential avenues leading to the enhancement of flexible ureteroscopy.

## Figures and Tables

**Figure 1 jcm-12-01093-f001:**
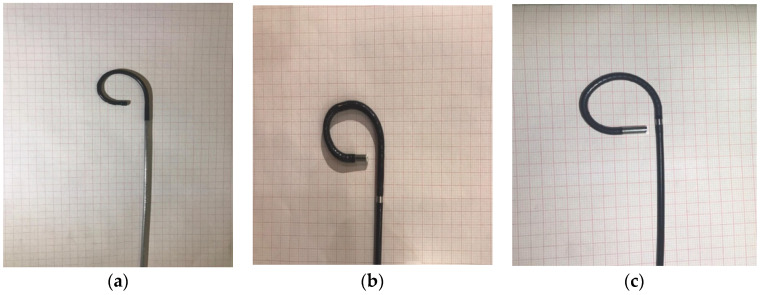
Flexible ureteroscopes USCOPE UE3022 (**a**), URF V (**b**), and URF V2 (**c**); Maximal deflection with empty working channel.

**Figure 2 jcm-12-01093-f002:**
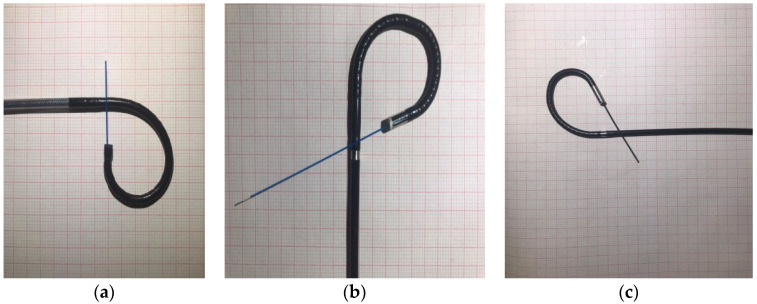
Flexible ureteroscopes USCOPE UE3022 (**a**), URF V (**b**), and URF V2 (**c**); Maximal tip deflection with laser fiber inserted through the working channel.

**Table 1 jcm-12-01093-t001:** Main technical features of the three studied models of flexible ureteroscopes (data provided by the manufacturers).

Model	Diameter (Tip/Tube)	Working Channel	Working Length	Deflection (Up/Down)	Field Of View	Direction of View
Olympus URF-V	8.5 F/9.9 F	1 × 3.6 F	670 mm	180°/275°	90°	0°
Olympus URF-V2	8.4 F	1 × 3.6 F	670 mm	275°/275°	80°	0°
Uscope UE3022	9.2 F	1 × 3.6 F	650 mm	270°/270°	120°	0°

**Table 2 jcm-12-01093-t002:** Comparative visibility and maneuverability scores of the three devices.

Model	Visibility Score	Maneuverability Score
Olympus URF-V	4.8	3.8
Olympus URF-V2	4.7	4.2
Uscope UE3022	4.8	4.2

**Table 3 jcm-12-01093-t003:** Influence of various accessory instruments over irrigation flow of the flexible ureteroscopes.

Type of Accessory Instruments	150 cm H_2_O Irrigation (mL/min)		
	URF-V	URF-V2	Uscope UE3022
None (empty channel)	22	21	35
0.035 Guidewire	1	1	2
Ho:YAG laser fiber	9	10	19
ZeroTip basket	1	0	2

**Table 4 jcm-12-01093-t004:** Influence of various accessory instruments over the deflection of the flexible ureteroscopes.

Type of Accessory Instruments	Maximal Deflection (One Direction/Second Direction)		
	URF-V	URF-V2	USCOPE UE3022
Manufacturer data	270°/180°	275°/275°	270°/270°
None (empty channel)	271°/172°	273°/274°	275°/269°
0.035 Guidewire	265°/157°	244°/240°	270°/269°
Ho:YAG laser fiber	249°/156°	238°/238°	270°/269°
ZeroTip basket	261°/155°	252°/248°	272°/260°

## Data Availability

This study did not report any data to the public data link.

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
