# Peer review of "Comparison of Flexible Ureteroscope Performance between Reusable and Single-Use Models"

_jcm, 2023, doi:10.3390/jcm12031093_

Round 1
Reviewer 1 Report
Authors compared two flexibles’ reusables ureteroscopes versus a single use one, both intraoperatively and in ex vivo settings.
The topic of the paper is interesting, but there are some issues that need to be clarified.
The authors talk about the high costs of the reusable ureteroscope without making a cost analysis compared to the single use one. At least an overview of the single one costs would be appreciated.
The considerations about the irrigation flow and the deflection angle are useful, but the subjective assessments in terms of visibility and maneuverability in vivo not so much.
Eventually an objective evaluation ex vivo of the parameters of visibility and maneuverability would be valuable.
The flexible ureteroscopes were compared by the same urologist. I wonder how many years of experience has the urologist who evaluated the ureteroscopes.
Author Response
Dear Reviewer 1,
First of all, we want to thank you for your observations!
In order to respond to your questions, we modified the present paper accordingly:
- The authors talk about the high costs of the reusable ureteroscope without making a cost analysis compared to the single use one. At least an overview of the single one costs would be appreciated.
Answer: Thank you very much for your suggestions. We truly believe that these suggestions will improve the value of our paper. We introduced the following phrase. in the Results section: "In regard to the acquisition costs, in our region, the reusable ureteroscope is 19 fold more expensive than the reusable one. The exploitation costs are more difficult to assess."
- The considerations about the irrigation flow and the deflection angle are useful, but the subjective assessments in terms of visibility and maneuverability in vivo not so much. Eventually an objective evaluation ex vivo of the parameters of visibility and maneuverability would be valuable.
Answer: You are right, there are certain problems associated with subjective assessments of various parameters, with potential bias sources involved. We tried to chose a design that will be as objective as possible. The assessment was made by a single experienced urologist to eliminate as many error factors as possible, however we know that this can be a bias generating factor by itself. Thank you for the valuable suggestion. We are exploring solutions to evaluate ex vivo the visibility and maneuverability for future studies. The following statement was added in the limitations section:"The fact that some parameters were subjectively evaluated may be another bias source.”
- The flexible ureteroscopes were compared by the same urologist. I wonder how many years of experience has the urologist who evaluated the ureteroscopes.
Answer: The interventions were performed by the same experienced surgeon, trying to make the subjective evaluation of visibility and maneuverability as unbiased as possible. The surgeon has 19 years of experience with flexible ureteroscopy, and has nowadays a rate of around 500 flexible ureteroscopies/year. We commented in the Material and Methods section: "The flexible ureteroscopes were compared intraoperatively by the same urologist, with great experience in the field of flexible ureteroscopy”.
Reviewer 2 Report
Dear Authors,
I read with interest your article entitled “Performance comparison of reusable versus single use models 2 of flexible ureteroscopes”.
In times of increasing attention towards ecology and environmental sustainability, the use of reusable model surely represents a hot topic. Below my comments:
ABSTRACT:
· You mention “performance and limitations”, please specify at least the most important characteristics you evaluated.
INTRODUCTION: no comments
MATERIAL AND METHODS:
· The order of the instruments in Table 1 does not match the order in the text. Please modify accordingly.
· When reporting the angles, please use “ ° “ instead of “ 0 “, since the circle is not a zero. Please correct it throughout the text (e.g., Table 3).
· Why did you place saline solution 150cm above the endoscope? This value is hardly representing the intraoperative conditions (I would say that 50-70cm is a real life setting).
· Were the fURS performed by a single surgeon? Which was his/her level of expertise? Please comment.
DISCUSSION
· In the limitations, I would add the fact that reusable ureteroscopes were not brand new at the time the study started.
· I suggest you to include this article in the discussion, since it would add an interest (and contrasting) point of view: Baboudjian M. et al., Life Cycle Assessment of Reusable and Disposable Cystoscopes: A Path to Greener Urological Procedures. Eur Urol Focus. 2022 Dec 20:S2405-4569(22)00291-7. doi: 10.1016/j.euf.2022.12.006.
Author Response
Dear Reviewer 2,
The reviewing process represents one of the most important parts and requires much time and involvement for scientific development.
Thank you for your time!
Accordingly, to your suggestions, we tried to respond as well as we could point by point as follows:
ABSTRACT:
- You mention “performance and limitations”, please specify at least the most important characteristics you evaluated.
Answer: Thank you for your suggestion. The abstract was modified accordingly: "Influence of different instruments over irrigation flow and deflection were assessed ex vivo. For the in vivo study, 20 consecutive patients underwent retrograde fURS using URF-V, 20 patients with URF-V2, and 20 patients with single-use fURS. During the procedures, visibility and maneuverability of each fURS were rated by the same urologist and compared."
INTRODUCTION: no comments
MATERIAL AND METHODS:
- The order of the instruments in Table 1 does not match the order in the text. Please modify accordingly.
Answer: You are totally right. We changed the order in the text so now it will match the order in all the Tables.
- When reporting the angles, please use “ ° “ instead of “ 0 “, since the circle is not a zero. Please correct it throughout the text (e.g., Table 3).
Answer: Thank you for the suggestion. We replaced the “0" with “o" everywhere in the text where angles were reported
- Why did you place saline solution 150cm above the endoscope? This value is hardly representing the intraoperative conditions (I would say that 50-70cm is a real life setting).
Answer: We know that there are variations regarding the height the saline solution is placed in various urological departments. Many of these variations where described and cited in the literature: 100 cm H2O (Dwayne C, Rustom PM, Konstantinos S et al. An Investigation of the Basic Physics of Irrigation in Urology and the Role of Automated Pump Irrigation in Cystoscopy, The Scientific World Journal, vol. 2012, Article ID 476759, 6 pages, 2012. https://doi.org/10.1100/2012/476759), 183 cmH2O (Hendlin K, Weiland D, Monga M. (2008). Impact of Irrigation Systems on Stone Migration. Journal of endourology / Endourological Society. 22. 453-8. 10.1089/end.2007.0260) etc. In our operating theatre the saline solution is placed 150 cm above the endoscope and we maintained this height also for the ex vivo evaluation. The following phrase from the Materials and methods section was changed in order to be more explanatory: "For the irrigation flow measurements, saline was placed at 150 cm above the endoscope, the same setting that we use in the operating room."
- Were the fURS performed by a single surgeon? Which was his/her level of expertise? Please comment.
Answer: The interventions were performed by the same experienced surgeon, trying to make the subjective evaluation of visibility and maneuverability as unbiased as possible. The surgeon has 19 years of experience with flexible ureteroscopy, and has nowadays a rate of around 500 flexible ureteroscopies/year. We commented in the Material and Methods section: "The flexible ureteroscopes were compared intraoperatively by the same urologist, with great experience in the field of flexible ureteroscopy”.
DISCUSSION
- In the limitations, I would add the fact that reusable ureteroscopes were not brand new at the time the study started.
Answer: You are perfectly right. We added the following statement where the limitations of the study were described: "This, along with the small sample size, the number of trials and the fact that reusable ureteroscopes were not brand new during the study are the limitations of the evaluation."
- I suggest you to include this article in the discussion, since it would add an interest (and contrasting) point of view: Baboudjian M. et al., Life Cycle Assessment of Reusable and Disposable Cystoscopes: A Path to Greener Urological Procedures. Eur Urol Focus. 2022 Dec 20:S2405-4569(22)00291-7. doi: 10.1016/j.euf.2022.12.006.
Answer: Thank you for your valuable suggestion. We added the following paragraph: "The impact on the environment of using different instruments and techniques should also be considered. By evaluating the disinfection reprocessing of reusable endoscopes with the complete lifespan of the single-use ones, the latter is associated with a reduction of at least 33% in the climate change category, 50% in the mineral resources' depletion category, 51% in the ecotoxicity category, 71% in the acidification category, and 49% in the eutrophication category according to a study by Baboudjian M. et al[31]”. We think that it will increase the value of our article’s discussions.
Reviewer 3 Report
Further investigation of single use scopes is very important. I am concerned that only Olympus scopes were used with no other comparators. The more common disposable ureteroscope is the lithovue in the US. Can you clarify why you chose that particular scope?
Author Response
Dear Reviewer 3,
Your comments were very well appreciated by the authors. We are aware that networking and involvement in the scientific process represent one of the major advantages of the modern era but also require a lot of time and effort.
Thank you for your observations.
We grouped our responses as follows:
- Further investigation of single use scopes is very important. I am concerned that only Olympus scopes were used with no other comparators. The more common disposable ureteroscope is the lithovue in the US. Can you clarify why you chose that particular scope?
Answer: We used in our study flexible ureteroscope models that are popular in our region. Also, the manufacturer of the single use scope is describing an irrigation flow superior to reusable scopes that have a similar 3.6F working channel and we wanted to evaluate this in an objective manner. To make things more explanatory we added the following statement in Material and Methods section: "We compared three of the latest models of flexible ureteroscopes which are popular in our region: two reusable digital (Olympus URF-V and Olympus URF-V2) versus the one single use (Pusen MedicalUscope UE3022).” The difference in irrigation is commented in the Discussion section.
Reviewer 4 Report
I revised manuscript number jcm-2170797 entitled “Performance comparison of reusable versus single use models of flexible ureteroscopes”.
In this study, the authors aimed to compare two of the latest flexible reusable ureteroscopes versus the single use one, both intraoperatively and in ex vivo settings, using objective as well as subjective parameters.
This paper is nicely written; however, some specifications need to be made:
Material and methods
- What kind of ureteral access sheath did you use? Please specify the size.
- Explain in detail the procedure you did intraoperatively, it is not clear. Size of the stone? Upper pole?
- Was the same operator the one that did all the tests?
Author Response
Dear Reviewer 4,
Your suggestions were very well received by the authors!
Thank you for your time and effort!
We answered each question separately and included the modifications in the original paper.
- What kind of ureteral access sheath did you use? Please specify the size.
Answer: Thank you for your suggestion. In all cases a Cook Flexor 10.7/12F access sheath was used. The following statement was added to the Materials and Methods section: "A Cook Flexor 10.7/12 F was used in all cases."
- Explain in detail the procedure you did intraoperatively, it is not clear. Size of the stone? Upper pole?
Answer: We truly believe that these additions will improve the value of our paper. The following paragraph was added to present in detail our technique: "Operative technique "The procedures were performed with the patient in standard lithotomy position, under spinal anesthesia. After cystoscopic identification of ureteral orifices, a 0.035" stiff hydrophilic guidewire is placed into the renal cavities under fluoroscopic control. A Cook Flexor 10.7/12 F was used in all cases. Initially, a thorough inspection of upper, middle and inferior calyx was performed. After we identified the stone, Holmium laser lithotripsy using a 270 µm fiber was performed in fragmentation mode, dusting or both. When necessary, baskets were used for stone retrieval. At the end of the procedure, another inspection of all the caliceal groups was performed. The urologist who performed the interventions was instructed to score the maneuvrability during the two inspections of all the caliceal groups, evaluating the success and also the ease in accessing all the calices. A JJ stent was indwelled at the end of the procedure.”. In this study we chose to evaluate consecutive cases, because the technical characteristics of the endoscopes were evaluated. The groups were not paired in regard to the stone characteristics and this is why the efficacy and complications rate were not compared. This was mentioned as a limitation of the study. In another situation, the fatiguability associated with multiple use of the reusable ureteroscope would introduce an unwanted bias in our analysis.
- Was the same operator the one that did all the tests?
Answer: The interventions were performed by the same experienced surgeon, trying to make the subjective evaluation of visibility and maneuverability as unbiased as possible. The surgeon has 19 years of experience with flexible ureteroscopy, and has nowadays a rate of around 500 flexible ureteroscopies/year. We commented in the Material and Methods section: "The flexible ureteroscopes were compared intraoperatively by the same urologist, with great experience in the field of flexible ureteroscopy”.
Reviewer 5 Report
1) In the irrigation flow measurement, what instruments did the Authors use?
2) I think the Authors should add a table which shows the main characteristics of the cohort of patients they used
3) How are the three groups? I think the Authors could add a table with the main characteristics and a homogeneity study
Author Response
Dear Reviewer 5,
Thank you for taking the time to review this article! Your comments showed involvement in this process and were appreciated by the authors.
Regarding the first observation, a flowmeter was used in the measurement process.
Unfortunately, this study did not include data on the patient's characteristics because it aims to evaluate the technical qualities of the tools used in daily activity.
Regarding the last observation, the authors did not consider it necessary to include a homogeneity study because consecutive cases were used in all 3 groups precisely to eliminate the risk of bias. The purpose of this study is to evaluate specific characteristics of flexible ureteroscopes used in daily use conditions.
Round 2
Reviewer 1 Report
Dear authors, thank you to respond to my previous comments.
You can enrich the discussion with the following papers:
Fluoroless versus conventional ureteroscopy for urinary stones: a systematic review and meta-analysis
Liao PENG, Wei WANG, Xiaoshuai GAO, Xingpeng DI, Deyi LUO *
Minerva Urology and Nephrology 2021 June;73(3):309-32
Intraoperative and postoperative surgical complications after ureteroscopy, retrograde intrarenal surgery, and percutaneous nephrolithotomy: a systematic review
Author Response
Dear Reviewer 1,
Thank you for your involvement in improving our paper!
Our team appreciates your efforts which significantly increased our paper quality.
The suggested papers were cited in the present article.
Intraoperative and postoperative surgical complications after ureteroscopy, retrograde intrarenal surgery, and percutaneous nephrolithotomy: a systematic review - citation nr. 3
Fluoroless versus conventional ureteroscopy for urinary stones: a systematic review and meta-analysis
Liao PENG, Wei WANG, Xiaoshuai GAO, Xingpeng DI, Deyi LUO *
Minerva Urology and Nephrology 2021 June;73(3):309-32 - citation nr. 33.
Reviewer 5 Report
You followed Reviewers' suggestions and now the manuscript is more clear and precise
Author Response
Dear Reviewer 5,
Thank you very much for your time and efforts!